# Bacterial contribution to genesis of the novel germ line determinant *oskar*

Leo Blondel[1], Tamsin EM Jones[2†], Cassandra G Extavour[1,2]*

[1]Department of Molecular and Cellular Biology, Harvard University, Cambridge, United States; [2]Department of Organismic and Evolutionary Biology, Harvard University, Cambridge, United States

**Abstract** New cellular functions and developmental processes can evolve by modifying existing genes or creating novel genes. Novel genes can arise not only via duplication or mutation but also by acquiring foreign DNA, also called horizontal gene transfer (HGT). Here we show that HGT likely contributed to the creation of a novel gene indispensable for reproduction in some insects. Long considered a novel gene with unknown origin, *oskar* has evolved to fulfil a crucial role in insect germ cell formation. Our analysis of over 100 insect Oskar sequences suggests that *oskar* arose *de novo via* fusion of eukaryotic and prokaryotic sequences. This work shows that highly unusual gene origin processes can give rise to novel genes that may facilitate evolution of novel developmental mechanisms.

*For correspondence:
extavour@oeb.harvard.edu

Present address: †European Bioinformatics Institute, EMBL-EBI, Wellcome Genome Campus, Hinxton, United Kingdom

Competing interests: The authors declare that no competing interests exist.

## Introduction

Heritable variation is the raw material of evolutionary change. Genetic variation can arise from mutation and gene duplication of existing genes (*Taylor and Raes, 2004*), or through *de novo* processes (*Tautz and Domazet-Lošo, 2011*), but the extent to which such novel, or 'orphan' genes participate significantly in the evolutionary process is unclear. Mutation of existing cis-regulatory (*Wittkopp and Kalay, 2012*) or protein coding regions (*Hoekstra and Coyne, 2007*) can drive evolutionary change in developmental processes. However, recent studies in animals and fungi suggest that novel genes can also drive phenotypic change (*Chen et al., 2013*). Although counterintuitive, novel genes may be integrating continuously into otherwise conserved gene networks, with a higher rate of partner acquisition than subtler variations on preexisting genes (*Zhang et al., 2015*). Moreover, in humans and fruit flies, a large proportion of novel genes are expressed in the brain, suggesting their participation in the evolution of major organ systems (*Zhang et al., 2012*; *Chen et al., 2012*). However, while next generation sequencing has improved their discovery, the developmental and evolutionary significance of novel genes remains understudied.

The mechanism of formation of a novel gene may have implications for its function. Novel genes that arise by duplication, thus possessing the same biophysical properties as their parent genes, have innate potential to participate in preexisting cellular and molecular mechanisms (*Taylor and Raes, 2004*). However, orphan genes lacking sequence similarity to existing genes must form novel functional molecular relationships with extant genes, in order to persist in the genome. When such genes arise by introduction of foreign DNA into a host genome through horizontal gene transfer (HGT), they may introduce novel, already functional sequence information into a genome. Whether genes created by HGT show a greater propensity to contribute to or enable novel processes is unclear. Endosymbionts in the host germ line cytoplasm (germ line symbionts) could increase the occurrence of evolutionarily relevant HGT events, as foreign DNA integrated into the germ line genome is transferred to the next generation. HGT from bacterial endosymbionts into insect genomes appears widespread, involving transfer of metabolic genes or even larger genomic

fragments to the host genome (see for example *Dunning Hotopp et al., 2007*; *Acuna et al., 2012*; *Sloan et al., 2014*; *Husnik et al., 2013*).

Here we examined the evolutionary origins of the *oskar* (*osk*) gene, long considered a novel gene that evolved to be indispensable for insect reproduction (*Lehmann, 2016*). First discovered in *Drosophila melanogaster* (*Lehmann and Nüsslein-Volhard, 1986*), *osk* is necessary and sufficient for assembly of germ plasm, a cytoplasmic determinant that specifies the germ line in the embryo. Germ plasm-based germ line specification appears derived within insects, confined to insects that undergo metamorphosis (Holometabola) (*Ewen-Campen et al., 2012*; *Extavour and Akam, 2003*). Initially thought exclusive to Diptera (flies and mosquitoes), its discovery in a wasp, another holometabolous insect with germ plasm (*Lynch et al., 2011*), led to the hypothesis that *oskar* originated as a novel gene at the base of the Holometabola approximately 300 Mya, facilitating the evolution of insect germ plasm as a novel developmental mechanism (*Lynch et al., 2011*). However, its subsequent discovery in a cricket (*Ewen-Campen et al., 2012*), a hemimetabolous insect without germ plasm (*Ewen-Campen et al., 2013*), implied that *osk* was instead at least 50 My older, and that its germ plasm role was derived rather than ancestral (*Abouheif, 2013*). Despite its orphan gene status, *osk* plays major developmental roles, interacting with the products of many genes highly conserved across animals (*Lehmann, 2016*; *Jeske et al., 2015*; *Jeske et al., 2017*). *osk* thus represents an example of a novel gene that not only functions within pre-existing gene networks in the nervous system (*Ewen-Campen et al., 2012*), but has also evolved into the only animal gene that has been experimentally demonstrated to be both necessary and sufficient to specify functional primordial germ line cells (*Ephrussi and Lehmann, 1992*; *Kim-Ha et al., 1991*).

The evolutionary origins of this remarkable gene are unknown. Osk contains two biophysically conserved domains, an N-terminal LOTUS domain and a C-terminal hydrolase-like domain called OSK (*Jeske et al., 2015*; *Yang et al., 2015*; *Figure 1a*). An initial BLASTp search using the full-length *D. melanogaster osk* sequence as a query yielded either other holometabolous insect *osk* genes, or partial hits for the LOTUS or OSK domains (E-value < 0.01; *Source data 1*: BLAST search results). This suggested that full length *osk* was unlikely to be a duplication of any other known gene. This prompted us to perform two more BLASTp searches, one using each of the two conserved Osk protein domains individually as query sequences. Strikingly, in this BLASTp search, although we recovered several eukaryotic hits for the LOTUS domain, we recovered no eukaryotic sequences that resembled the OSK domain, even with very low E-value stringency (E-value < 10; see Materials and methods section "*BLAST searches of oskar*" for an explanation of E-value threshold choices; *Source data 1*: BLAST search results).

To understand this anomaly, we built an alignment of 95 Oskar sequences (*Source data 1* Alignments>OSKAR_MUSCLE_FINAL.fasta; *Supplementary file 1A and B*) and used a custom iterative HMMER sliding window search tool to compare each domain with protein sequences from all domains of life. Sequences most similar to the LOTUS domain were almost exclusively eukaryotic sequences (*Supplementary file 1C*). In contrast, those most similar to the OSK domain were bacterial, specifically sequences similar to SGNH-like hydrolases (*Jeske et al., 2015*; *Yang et al., 2015*) (Pfam Clan: SGNH_hydrolase - CL0264; *Supplementary file 1D*; *Figure 1b*). To visualize their relationships, we graphed the sequence similarity network for the sequences of these domains and their closest hits. We observed that the majority of LOTUS domain sequences clustered within eukaryotic sequences (*Figure 1c*). In contrast, OSK domain sequences formed an isolated cluster, a small subset of which formed a connection to bacterial sequences (*Figure 1d*). These data are consistent with a previous suggestion, based on BLAST results (*Lynch et al., 2011*), that HGT from a bacterium into an ancestral insect genome may have contributed to the evolution of *osk*. However, this possibility was not formally addressed by previous analyses, which were based on alignments of full length Osk containing only eukaryotic sequences as outgroups (*Ewen-Campen et al., 2012*). To rigorously test this hypothesis, we therefore performed phylogenetic analyses of the two domains independently. A finding that LOTUS sequences were nested within eukaryotes, while OSK sequences were nested within bacteria, would provide support for the HGT hypothesis.

Both Maximum likelihood and Bayesian approaches confirmed this prediction (*Figure 2a*, *Figure 2—figure supplements 1* and *2*), and these results were robust to changes in the methods of sequence alignment (*Figure 2—figure supplements 6*, *7*, *8*, *9*, *10*). As expected, LOTUS sequences from Osk proteins were related to other eukaryotic LOTUS domains, to the exclusion of the only three bacterial sequences that met our E-value cutoff for inclusion in the analyses (*Figure 2a*,

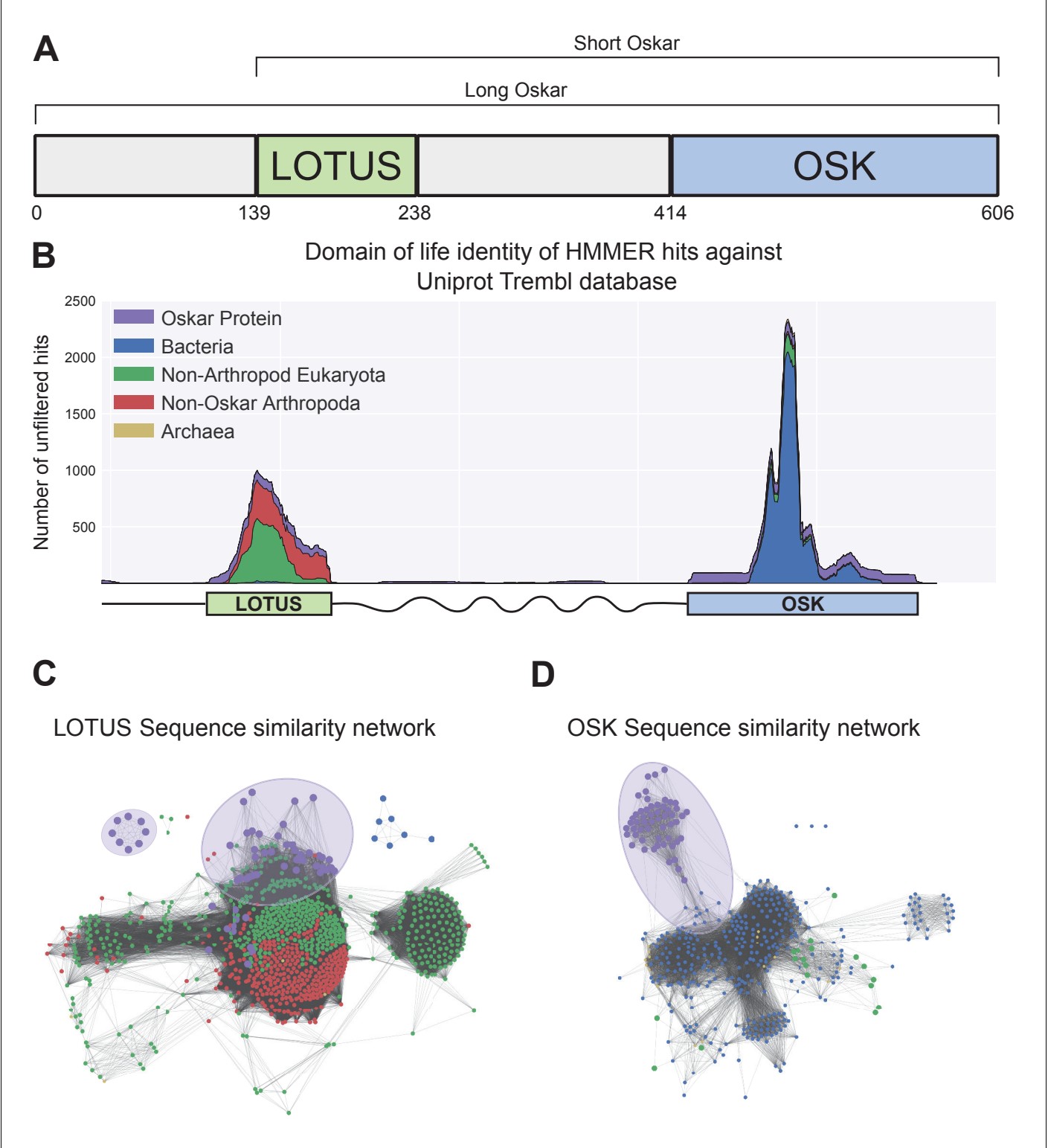

**Figure 1.** Sequence analysis of the Oskar gene. (a) Schematic representation of the Oskar gene. The LOTUS and OSK hydrolase-like domains are separated by a poorly conserved region of predicted high disorder and variable length between species. In some dipterans, a region 5′ to the LOTUS domain is translated to yield a second isoform, called Long Oskar. Residue numbers correspond to the *D. melanogaster* Osk sequence. (b) Stackplot of domain of life identity of HMMER hits across the protein sequence. For a sliding window of 60 Amino Acids across the protein sequence (X axis), the number of hits in the Trembl (UniProt) database (Y axis) is represented and color coded by domain of life origin (see Materials and methods: Iterative HMMER search of OSK and LOTUS domains), stacked on top of each other. (c, d) EFI-EST-generated graphs of the sequence similarity network of the

*Figure 1 continued on next page*

*Figure 1 continued*

LOTUS (**c**) and OSK (**d**) domains of Oskar (*Gerlt et al., 2015*). Sequences were obtained using HMMER against the UniProtKB database. Most Oskar LOTUS sequences cluster within eukaryotes and arthropods. In contrast, Oskar OSK sequences cluster most strongly with a small subset of bacterial sequences.

*Figure 2—figure supplements 1* and *2*; see Materials and methods and Supplemental Text). LOTUS sequences from non-Oskar proteins were almost exclusively eukaryotic. (*Supplementary file 1*); only three bacterial sequences matched the LOTUS domain with an E-value < 0.01. Osk LOTUS domains clustered into two distinct clades, one comprising all Dipteran sequences, and the other comprising all other Osk LOTUS domains examined from both holometabolous and hemimetabolous orders (*Figure 2a*). Dipteran Osk LOTUS sequences formed a monophyletic group that branched sister to a clade of LOTUS domains from Tud5 family proteins of non-arthropod animals (NAA). NAA LOTUS domains from Tud7 family members were polyphyletic, but most of them formed a clade branching sister to (Osk LOTUS + NAA Tud5 LOTUS). Non-Dipteran Osk LOTUS domains formed a monophyletic group that was related in a polytomy to the aforementioned (NAA Tud7 LOTUS + (Dipteran Osk LOTUS + NAA Tud5 LOTUS)) clade, and to various arthropod Tud7 family LOTUS domains.

The fact that Tud7 LOTUS domains are polyphyletic suggests that arthropod domains in this family may have evolved differently than their homologues in other animals. The relationships of Dipteran LOTUS sequences were consistent with the current hypothesis for interrelationships between Dipteran species (*Kirk-Spriggs and Sinclair, 2017*). Similarly, among the non-Dipteran Osk LOTUS sequences, the hymenopteran sequences form a clade to the exclusion of the single hemimetabolous sequence (from the cricket *Gryllus bimaculatus*), consistent with the monophyly of Hymenoptera (*Peters et al., 2017*). It is unclear why Dipteran Osk LOTUS domains cluster separately from those of other insect Osk proteins. We speculate that the evolution of the Long Oskar domain (*Vanzo and Ephrussi, 2002*; *Hurd et al., 2016*), which appears to be a novelty within Diptera (*Source data 1*: Alignments>OSKAR_MUSCLE_FINAL.fasta), may have influenced the evolution of the Osk LOTUS domain in at least some of these insects. Consistent with this hypothesis, of the 17 Dipteran *oskar* genes we examined, the seven *oskar* genes possessing a Long Osk domain clustered into two clades based on the sequences of their LOTUS domain. One of these clades comprised five *Drosophila* species (*D. willistoni*, *D. mojavensis*, *D. virilis*, *D. grimshawi* and *D. immigrans*), and the second was composed of two calyptrate flies from different superfamilies, *Musca domestica* (Muscoidea) and *Lucilia cuprina* (Oestroidea).

In summary, the LOTUS domain of Osk proteins is most closely related to a number of other LOTUS domains found in eukaryotic proteins, as would be expected for a gene of animal origin, and the phylogenetic interrelationships of these sequences are largely consistent with the current species or family level trees for the corresponding insects.

In contrast, OSK domain sequences were nested within bacterial sequences (*Figure 2b*, *Figure 2—figure supplements 3* and *4*). This bacterial, rather than eukaryotic, affinity of the OSK domain was recovered even when different sequence alignment methods were used (*Figure 2—figure supplements 7*, *8*, *9*, *10* and *11*). The only eukaryotic proteins emerging from the iterative HMMER search for OSK domain sequences that had an E-value < 0.01 were all from fungi. All five of these sequences were annotated as Carbohydrate Active Enzyme 3 (CAZ3), and all CAZ3 sequences formed a clade that was sister to a clade of primarily Firmicutes. Most bacterial sequences used in this analysis were annotated as lipases and hydrolases, with a high representation of GDSL-like hydrolases (*Supplementary file 1D*). OSK sequences formed a monophyletic group but did not branch sister to the other eukaryotic sequences in the analysis. Within this OSK clade, the topology of sequence relationships was largely concordant with the species tree for insects (*Misof et al., 2014*), as we recovered monophyletic Diptera to the exclusion of other insect species. However, the single orthopteran OSK sequence (from the cricket *G. bimaculatus*) grouped within the Hymenoptera, rather than branching as sister to all other insect sequences in the tree, as would be expected for this hemimetabolous sequence (*Misof et al., 2014*).

Importantly, OSK sequences did not simply form an outgroup to bacterial sequences. To formally reject the possibility that the eukaryotic OSK clade has a sister group relationship to all bacterial sequences in the analysis, we performed topology constraint analyses using the Swofford–Olsen–Waddell–Hillis (SOWH) test, which assigns statistical support to alternative phylogenetic topologies

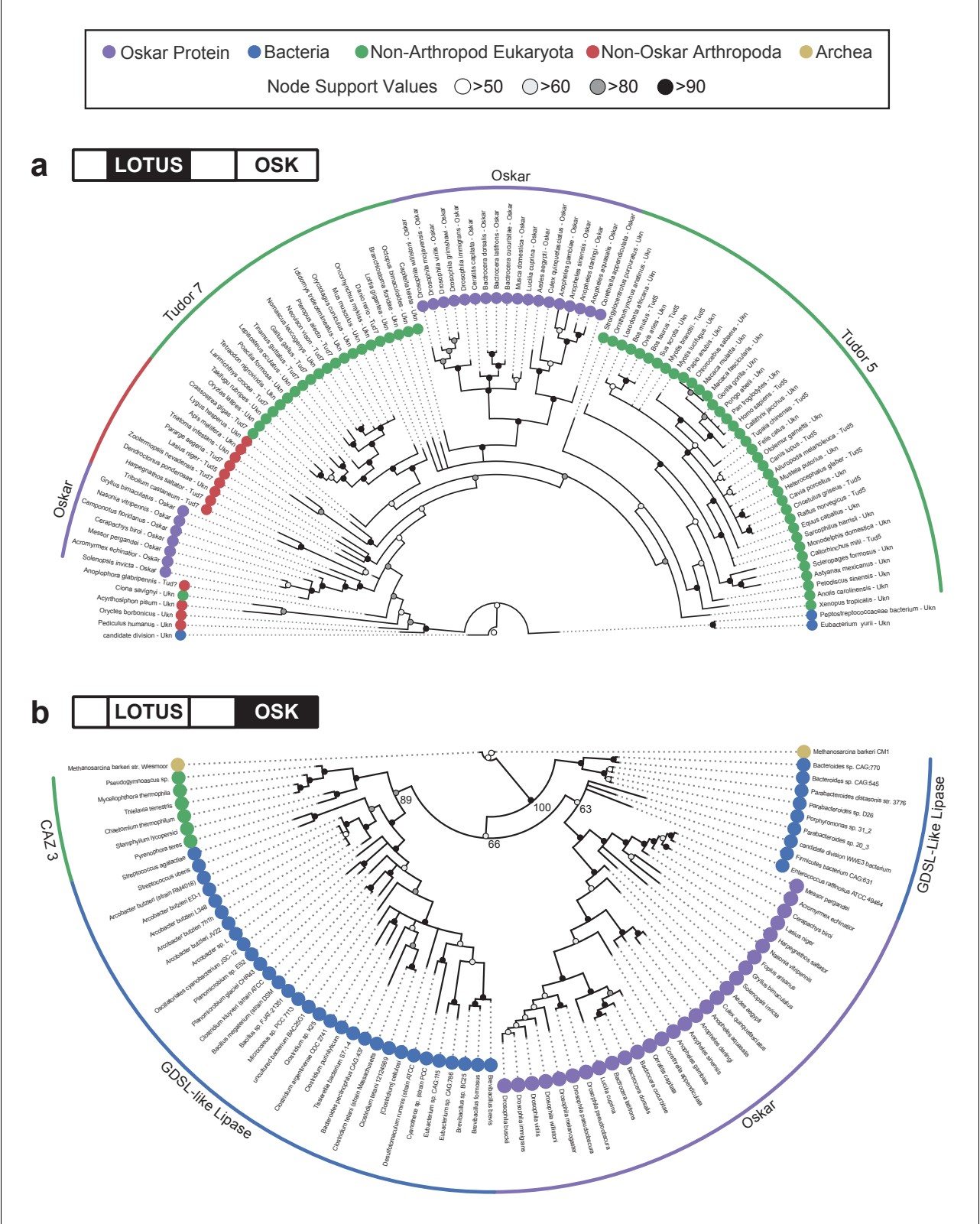

**Figure 2.** Phylogenetic analysis of the LOTUS and OSK domains. (**a**) Bayesian consensus tree for the LOTUS domain. Three major LOTUS-containing protein families are represented within the tree: Tudor 5, Tudor 7, and Oskar. Oskar LOTUS domains form two clades, one containing only dipterans and one containing all other represented insects (hymenopterans and orthopterans). The tree was rooted to the three bacterial sequences added in the dataset. (**b**) Bayesian consensus tree for the OSK domain. The OSK domain is nested within GDSL-like domains of bacterial species from phyla known

*Figure 2 continued on next page*

*Figure 2 continued*

to contain germ line symbionts in insects. The ten non-Oskar eukaryotic sequences in the analysis form a single clade comprising fungal Carbohydrate Active Enzyme 3 (CAZ3) proteins. For Bayesian and RaxML trees with all accession numbers and node support values see ***Figure 2—figure supplements 1–4***.

The online version of this article includes the following figure supplement(s) for figure 2:

**Figure supplement 1.** LOTUS Domain RaxML MUSCLE Tree.
**Figure supplement 2.** LOTUS Domain Bayesian MUSCLE Tree.
**Figure supplement 3.** OSK Domain RaxML MUSCLE Tree.
**Figure supplement 4.** OSK Domain Bayesian MUSCLE Tree.
**Figure supplement 5.** SOWHAT constrained trees and results.
**Figure supplement 6.** LOTUS Domain RaxML PRANK Tree.
**Figure supplement 7.** OSK Domain RaxML PRANK Tree.
**Figure supplement 8.** OSK Tree PRANK Comparison.
**Figure supplement 9.** LOTUS Tree PRANK Comparison.
**Figure supplement 10.** LOTUS Domain RaxML T-Coffee Tree.
**Figure supplement 11.** OSK Domain RaxML T-Coffee Tree.
**Figure supplement 12.** OSK Tree T-Coffee Comparison.
**Figure supplement 13.** LOTUS Tree T-Coffee Comparison.

(*Swofford et al., 1996*). We used the SOWHAT tool (*Church et al., 2015*) to compare the HGT-supporting topology to two alternative topologies with constraints more consistent with vertical inheritance. The first was constrained by domain of life, disallowing paraphyletic relationships between sequences from the same domain of life (***Figure 2—figure supplement 5a***). The second required monophyly of Eukaryota but allowed paraphyletic relationships between bacterial and archaeal sequences (***Figure 2—figure supplement 5b***). We found that the topologies of both of these constrained trees were significantly worse than the result we had recovered with our phylogenetic analysis (***Figure 2—figure supplement 5***), namely that the closest relatives of the OSK domain were bacterial rather than eukaryotic sequences ***Figure 2b***, ***Figure 2—figure supplements 3*** and ***4***).

OSK sequences formed a well-supported clade nested within bacterial GDSL-like lipase sequences. The majority of these bacterial sequences were from the Firmicutes, a bacterial phylum known to include insect germ line symbionts (*Wheeler et al., 2013*; *Chepkemoi et al., 2017*). All other sequences from classified bacterial species, including a clade branching as sister to all other sequences, belonged either to the Bacteroidetes or to the Proteobacteria. Members of both of these phyla are also known germ line symbionts of insects (*Dunning Hotopp et al., 2007*; *Zchori-Fein et al., 2004*) and other arthropods (*Zchori-Fein and Perlman, 2004*). In sum, the distinct phylogenetic relationships of the two domains of Oskar are consistent with a bacterial origin for the OSK domain. Further, the specific bacterial clades close to OSK suggest that an ancient arthropod germ line endosymbiont could have been the source of a GDSL-like sequence that was transferred into an ancestral insect genome, and ultimately gave rise to the OSK domain of *oskar* (***Figure 3***).

While multiple mechanisms can give rise to novel genes, HGT is arguably among the least well understood, as it involves multiple genomes and ancient biotic interactions between donor and host organisms that are often difficult to reconstruct. In the case of *oskar*, however, the fact that both germ line symbionts (*Bourtzis and Miller, 2006*) and HGT events (*Dunning Hotopp et al., 2007*) are widespread in insects, provides a plausible biological mechanism consistent with our hypothesis that fusion of eukaryotic and bacterial domain sequences led to the birth of this novel gene. Under this hypothesis, this fusion would have taken place before the major diversification of insects, nearly 500 million years ago (*Misof et al., 2014*).

Once arisen, novel genes might be expected to disappear rapidly, given that pre-existing gene regulatory networks operated successfully without them (*Taylor and Raes, 2004*). However, it is clear that novel genes can evolve functional connections with existing networks, become essential (*Chen et al., 2010*), and in some cases lead to new functions (*Cornelis et al., 2012*) and contribute to phenotypic diversity (*Chen et al., 2013*). Even given the growing number of convincing examples of HGT from both prokaryotic and eukaryotic origins (see for example *Husnik and McCutcheon, 2018*; *Di Lelio et al., 2019*; *Wybouw et al., 2016*; *Quispe-Huamanquispe et al., 2017*), some authors suspect that the contribution of horizontal gene transfer to the acquisition of novel traits has

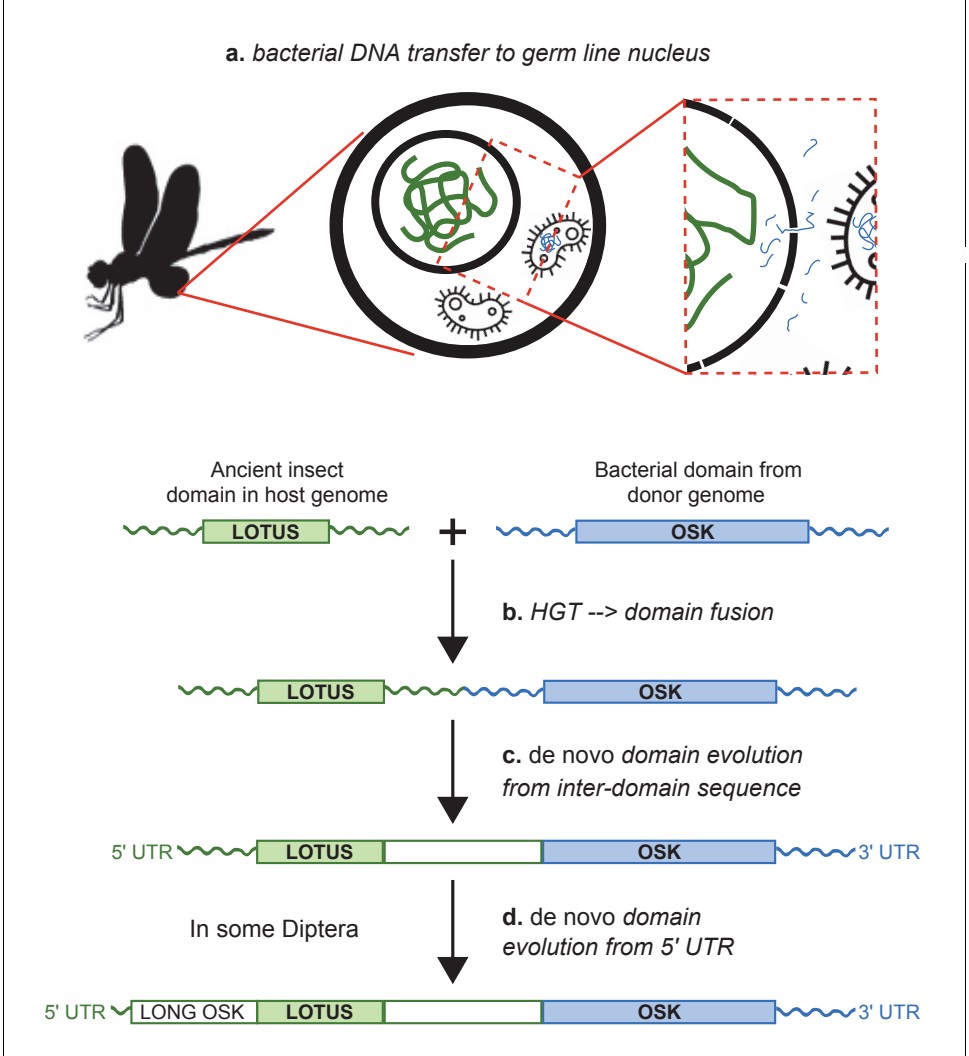

**Figure 3.** Hypothesis for the origin of *oskar*. Integration of the OSK domain close to a LOTUS domain in an ancestral insect genome. (a) DNA containing a GDSL-like domain from an endosymbiotic germ line bacterium is transferred to the nucleus of a germ cell in an insect common ancestor. (b) DNA damage or transposable element activity induces an integration event in the host genome, close to a pre-existing LOTUS-like domain. (c) The region between the two domains undergoes *de novo* coding evolution, creating an open reading frame with a unique, chimeric domain structure. (d) In some Diptera, including *D. melanogaster*, part of the 5' UTR of *oskar* has undergone *de novo* coding evolution to form the Long Oskar domain.

been underestimated across animals (*Boto, 2014*). Moreover, the functional contribution of genes horizontally transferred specifically from bacteria to insects has been documented for a range of adaptive phenotypes (see for example *Wilson and Duncan, 2015*; *López-Madrigal and Gil, 2017*; *Provorov and Onishchuk, 2018*), including digestive metabolism (*Acuna et al., 2012*; *Sloan et al., 2014*; *Shelomi et al., 2016*), glycolysis (*Zeng et al., 2018*) complex symbiosis (*Husnik et al., 2013*) and endosymbiont cell wall construction (*Bublitz et al., 2019*). *oskar* plays multiple critical roles in insect development, from neural patterning (*Ewen-Campen et al., 2012*; *Xu et al., 2013*) to oogenesis (*Jenny et al., 2006*). In the Holometabola, a clade of nearly one million extant species (*Rees and Cranston, 2017*), *oskar*'s co-option to become necessary and sufficient for germ plasm assembly is likely the cell biological mechanism underlying the evolution of this derived mode of insect germ line specification (*Ewen-Campen et al., 2012*; *Lynch et al., 2011*; *Abouheif, 2013*). Our study thus provides evidence that HGT can not only introduce functional genes into a host

genome, but also, by contributing sequences of individual domains, generate genes with entirely novel domain structures that may facilitate the evolution of novel developmental mechanisms.

## Materials and methods

### BLAST searches of Oskar

All BLAST (*Altschul et al., 1990*) searches were performed using the NCBI BLASTp tool suite on the non-redundant (nr) database. Amino Acid (AA) sequences of *D. melanogaster* full length Oskar (EMBL ID AAF54306.1), as well as the AA sequences for the *D. melanogaster* Oskar LOTUS (AA 139–238) and OSK (AA 414–606) domains were used for the BLAST searches. We used the default NCBI cut-off parameters (E-value cut-off of 10) for searches using OSK and LOTUS as queries, and a more stringent E-value threshhold of 0.01 for the search using full length *D. melanogaster* Oskar as a query. We chose an E-value threshold of 10 for LOTUS and OSK to capture potentially highly divergent homologs of the two domains, especially for the OSK domain, where we were looking for any viable candidate for a homologous eukaryotic domain. All BLAST searches results are included in the *Source data 1*: BLAST search results.

### Hidden Markov Model (HMM) generation and alignments of the OSK and LOTUS domains

101 1KITE transcriptomes (*Misof et al., 2014*; *Supplementary file 1A*) were downloaded and searched using the local BLAST program (BLAST+) using the tblastn algorithm with default parameters, with Oskar protein sequences of *Drosophila melanogaster, Aedes aegypti, Nasonia vitripennis* and *Gryllus bimaculatus* as queries (EntrezIDs: NP_731295.1, ABC41128.1, NP_001234884.1 and AFV31610.1 respectively). For all of these 1KITE transcriptome searches, predicted protein sequences from transcript data were obtained by in silico translation using the online ExPASy translate tool (https://web.expasy.org/translate/), taking the longest open reading frame. Publicly available sequences in the non-redundant (nr), TSA databases at NCBI, and a then-unpublished transcriptome (*Benton et al., 2016*) (kind gift of Matthew Benton and Siegfried Roth, University of Cologne) were subsequently searched using the web-based BLAST tool hosted at NCBI, using the tblastn algorithm with default parameters. Sequences used for queries were the four Oskar proteins described above, and newfound *oskar* sequences from the 1KITE transcriptomes of *Baetis pumilis, Cryptocercus wright,* and *Frankliniella cephalica*. For both searches, *oskar* orthologs were identified by the presence of BLAST hits on the same transcript to both the LOTUS (N-terminal) and OSK (C-terminal) regions of any of the query *oskar* sequences, regardless of E-values. The sequences found were aligned using MUSCLE (eight iterations) (*Edgar, 2004*) into a 46-sequence alignment (*Source data 1*: Alignments > OSKAR_MUSCLE_INITIAL.fasta). From this alignment, the LOTUS and OSK domains were extracted (*Source data 1*: Alignments > LOTUS_MUSCLE_INITIAL.fasta and Alignments > OSK_MUSCLE_INITIAL.fasta) to define the initial Hidden Markov Models (HMM) using the hmmbuild tool from the HMMER tool suite with default parameters (http://hmmer.org/; *Eddy, 2011*). 126 insect genomes and 128 insect transcriptomes (from the Transcriptome Shotgun Assembly TSA database: https://www.ncbi.nlm.nih.gov/Traces/wgs/?view=TSA) were subsequently downloaded from NCBI (download date September 29, 2015; *Supplementary file 1A*). Genomes were submitted to Augustus v2.5.5 (*Stanke et al., 2004*) (using the *D. melanogaster* exon HMM predictor) and SNAP v2006-07-28 (*Korf, 2004*) (using the default 'fly' HMM) for gene discovery. The resulting nucleotide sequence database comprising all 309 downloaded and annotated genomes and transcriptomes, was then translated in six frames to generate a non-redundant amino acid database (where all sequences with the same amino acid content are merged into one). This process was automated using a series of custom scripts available here: https://github.com/Xqua/Genomes. The non-redundant amino acid database was searched using the HMMER v3.1 tool suite (*Eddy, 2011*) and the HMM for the LOTUS and OSK domains described above. A hit was considered positive if it consisted of a contiguous sequence containing both a LOTUS domain and an OSK domain, with the two domains separated by an inter-domain sequence. We imposed no length, alignment or conservation criteria on the inter-domain sequence, as this is a rapidly-evolving region of Oskar protein with predicted high disorder (*Jeske et al., 2015*; *Yang et al., 2015*; *Ahuja and Extavour, 2014*). Positive hits were manually curated and added to the main alignment, and the search was performed

iteratively until no more new sequences meeting the above criteria were discovered. This resulted in a total of 95 Oskar protein sequences, (see *Supplementary file 1B* for the complete list). Using the final resulting alignment (*Source data 1*: Alignments > OSKAR_MUSCLE_FINAL.fasta), the LOTUS and OSK domains were extracted from these sequences (*Source data 1*: Alignments > LOTUS_MU-SCLE_FINAL.fasta and Alignments > OSK_MUSCLE_FINAL.fasta), and the final three HMM (for full-length Oskar, OSK, and LOTUS domains) used in subsequent analyses were created using hmmbuild with default parameters (*Source data 1*: HMM >OSK.hmm, HMM >LOTUS.hmm and HMM >OSKAR.hmm).

## Iterative HMMER search of OSK and LOTUS domains

A reduced version of TrEMBL (*U Consortium, 2005*) (v2016-06) was created by concatenating all hits (regardless of E-value) for sequences of the LOTUS domain, the OSK domain and full-length Oskar, using hmmsearch with default parameters and the HMM models created above from the final alignment. This reduced database was created to reduce potential false positive results that might result from the limited size of the sliding window used in the search approach described here. The full-length Oskar alignment of 1133 amino acids (*Source data 1*: Alignments > OSKAR_MUSCLE_FINAL.fasta) was split into 934 sub-alignments of 60 amino acids each using a sliding window of one amino acid. Each alignment was converted into a HMM using hmmbuild, and searched against the reduced TrEMBL database using hmmsearch using default parameters. Domain of life origin of every hit sequence at each position was recorded. Eukaryotic sequences were further classified as Oskar/Non-Oskar and Arthropod/Non-Arthropod. Finally, for the whole alignment, the counts for each category were saved and plotted in a stack plot representing the proportion of sequences from each category to create *Figure 1b*. The python code used for this search is available at https://github.com/Xqua/Iterative-HMMER.

## Sequence similarity networks

LOTUS and OSK domain sequences from the final alignment obtained as described above (see '*Hidden Markov Model (HMM) generation and alignments of the OSK and LOTUS domains*'; *Source data 1*: Alignments > LOTUS_MUSCLE_FINAL.fasta and Alignments > OSK_MUSCLE_FINAL.fasta) were searched against TrEMBL (*U Consortium, 2005*) (v2016-06) using HMMER. All hits with E-value <0.01 were consolidated into a fasta file that was then entered into the EFI-EST tool (*Gerlt et al., 2015*) using default parameters to generate a sequence similarity network. An alignment score corresponding to 30% sequence identity was chosen for the generation of the final sequence similarity network. Finally, the network was graphed using Cytoscape 3 (*Shannon et al., 2003*).

## Phylogenetic analysis based on MUSCLE alignment

For both the LOTUS and OSK domains, in cases where more than one sequence from the same organism was retrieved by the search described above in '*Iterative HMMER Search of OSK and LOTUS domains*', only the sequence with the lowest E-value was used for phylogenetic analysis. For the LOTUS domain, the first 97 best hits (lowest E-value) were selected, and the only three bacterial sequences that satisfied an E-value <0.01 were manually added. For *oskar* sequences, if more than one sequence per species was obtained by the search, only the single sequence per species with the lowest E-value was kept for analysis, generating a set of 100 sequences for the LOTUS domain, and 87 sequences for the OSK domain. Unique identifiers for all sequences used to generate alignments for phylogenetic analysis are available in *Supplementary files 1C, 1D*. For both datasets, the sequences were then aligned using MUSCLE (*Edgar, 2004*) (eight iterations) and trimmed using trimAl (*Capella-Gutiérrez et al., 2009*) with 70% occupancy. The resulting alignments that were subject to phylogenetic analysis are available in *Source data 1*: Alignments > LOTUS_MUSCLE_TREE.fasta and Alignments > OSK_MUSCLE_TREE.fasta. For the maximum likelihood tree, we used RaxML v8.2.4 (*Stamatakis, 2014*) with 1000 bootstraps, and the models were selected using the automatic RaxML model selection tool. The substitution model chosen for both domains was LGF. For the Bayesian tree inference, we used MrBayes V3.2.6 (*Huelsenbeck and Ronquist, 2001*) with a Mixed model (prset aamodel = Mixed) and a gamma distribution (lset rates = Gamma). We ran the Monte-Carlo for 4 million generations (std <0.01) for the OSK domain, and for 3 million generations

(std <0.01) for the LOTUS domain. For the tree comparisons (*Figure 2—figure supplements 8*, *9*), the RaxML best tree output from the MUSCLE and PRANK alignments were compared using the tool Phylo.io (*Robinson et al., 2016*).

## Phylogenetic analysis based on PRANK alignment

For the OSK domain, the raw full length sequences obtained from the HMMER search were aligned to each other using the HMMER HMM-based alignment tool: hmmalign, with the same HMM used to do the search, namely OSK.hmm (supplementary data: Data/HMM/OSK.hmm). Starting from this base alignment, we used the default alignment method option offered by PRANK (version: v.170427) (*Löytynoja, 2014*). We then used PRANK to realign those sequences, which in turn led to a usable alignment for phylogenetic analysis. This alignment was trimmed using the same parameters as described in *Hidden Markov Model (HMM) generation and alignments of the OSK and LOTUS domains* above. The final alignment is available in supplementary data: Alignment/OSK_prank_aligned.fasta. We then performed a phylogenetic analysis of this alignment using RAXML with the same parameters described in *Phylogenetic Analysis Based on MUSCLE Alignment* above. The resulting tree is presented in *Figure 2—figure supplements 7* and *8*.

For the LOTUS domain, the raw full length sequences obtained from the HMMER search were aligned to each other using the HMMER HMM-based alignment tool: hmmalign, with the same HMM used to do the search, namely LOTUS.hmm (Supplementary data: Data/HMM/LOTUS.hmm). Starting from this base alignment, we then used PRANK with default options to realign those sequences. This alignment was trimmed using the same parameters as described in the *Hidden Markov Model* (*HMM*) *generation and alignments of the OSK and LOTUS domains*. The final alignment is available in supplementary data: Alignments/LOTUS_prank_aligned.fasta. We then performed a phylogenetic analysis using RAXML with the same parameters described above in *Phylogenetic Analysis Based on MUSCLE alignment*. The resulting trees are presented in *Figure 2—figure supplements 6* and *9*.

## Phylogenetic analysis based on T coffee alignment

For the LOTUS and OSK domains, the raw full length sequences obtained from the HMMER search were aligned to each other using T-Coffee with its default parameters (*Notredame et al., 2000*). This alignment was trimmed using the same parameters as described in *Hidden Markov Model* (*HMM*) *generation and alignments of the OSK and LOTUS domains* above. The final alignment is available in supplementary data: Alignment/LOTUS_tcoffee_aligned.fasta Alignment/OSK_tcoffee_aligned.fasta. We then performed a phylogenetic analysis of this alignment using RAXML with the same parameters described in *Phylogenetic Analysis Based on MUSCLE Alignment* above. The resulting trees are presented in *Figure 2—figure supplements 10* and *11*.

## Visual comparison of phylogenetic trees

To compare the trees obtained with different alignment tools, we used Phylo.io (*Robinson et al., 2016*). The trees were imported in Newick format, and the Phylo.io tool generated the mirrored and aligned versions of the trees represented in *Figure 2—figure supplements 8*, *9*, *12* and *13*. The color of the branches is the tree similarity score, where lighter colors represent a higher number of topological differences. It is a custom implementation of the Jacard Index by Phylo.io.

## Statistical analysis of tree topology

To statistically evaluate our best-supported topology of the OSK and LOTUS trees, we compared constrained topologies to the highest likelihood trees using the SOWHAT tool (*Church et al., 2015*). SOWHAT automates the stringent SOWH phylogenetic topology test (*Swofford et al., 1996*), and compares the log likelihood between generated trees. We defined three constrained trees to test our results, one requiring monophyly of all domains of life, a second requiring only eukaryotic monophyly, and the last one requiring monophyly of the Oskar LOTUS domain (*Source data 1*: Data > Trees > constrained_kingdom_tree.tre, constrained_eukmono_tree.tre and constrained_lotus_mono_tree.tre). We then ran SOWHAT using its default parameters, 1000 bootstraps, and the two constrained trees against the OSK or LOTUS alignment used to generate the phylogenetic trees

(*Source data 1*: Alignments > OSK_MUSCLE_TREE.fasta and LOTUS_MUSCLE_TREE.fasta). All best trees generated by SOWHAT are available in (*Source data 1*: Data > Trees > SOWHAT_*_test.tre).

## Code availability

All custom code generated for this study is available in the GitHub repository https://github.com/extavourlab/Oskar_HGT, commit ID 6f6c4c50dfb9391567d70f9eea922f3876a4e153 (*Blondel et al., 2020*; copy archived at https://github.com/elifesciences-publications/Oskar_HGT).

## Scripts

All scripts used herein are hosted on GitHub at https://github.com/extavourlab/Oskar_HGT.

## Acknowledgements

We thank Sean Eddy, Chuck Davis, and Extavour lab members for discussion.

## Additional information

### Funding

| Funder | Author |
|---|---|
| Harvard University | Leo Blondel<br>Cassandra G Extavour<br>Tamsin E M Jones |

The funders had no role in study design, data collection and interpretation, or the decision to submit the work for publication.

### Author contributions

Leo Blondel, Data curation, Formal analysis, Validation, Visualization, Methodology, Writing - original draft, Writing - review and editing; Tamsin EM Jones, Data curation, Writing - review and editing; Cassandra G Extavour, Conceptualization, Supervision, Funding acquisition, Writing - original draft, Project administration, Writing - review and editing

### Author ORCIDs

Leo Blondel ⬤ http://orcid.org/0000-0003-2276-4821
Tamsin EM Jones ⬤ https://orcid.org/0000-0002-0027-0858
Cassandra G Extavour ⬤ https://orcid.org/0000-0003-2922-5855

### Decision letter and Author response

Decision letter https://doi.org/10.7554/eLife.45539.sa1
Author response https://doi.org/10.7554/eLife.45539.sa2

## Additional files

### Supplementary files

• Source data 1. Alignment and Sequence Classification Tools & Data. **Subfolder "Alignments"**: All sequences identified and analyzed in this study, in FASTA format and with corresponding Alignments. Subfolder BLAST search results: Results of BLASTP searches with full length Oskar, OSK or LOTUS domains as queries. **Subfolder "Data"**: Necessary files for running the different IPython notebooks: *a. Subfolder "HMM"*: HMM models used for iterative searching for sequences similar to full-length Oskar, LOTUS and OSK domains; *b. Subfolder "Taxonomy"*: Conversion table for Uni-Prot ID to taxon information (uniprot_ID_taxa.tsv); *c. Subfolder "Trees"*: Contains the tree files obtained from i. RaxML phylogenetic analyses of the OSK and LOTUS domains aligned with MUS-CLE, T-Coffee or PRANK; ii. MrBayes phylogenetic analyses of the OSK and LOTUS domains aligned with MUSCLE; iii. SOWHAT analyses.

- Supplementary file 1. Supplementary tables. (**A**) List of genomes and transcriptomes used for automated *oskar* search. (**B**) List of Oskar sequences used in the final alignment. (**C**) List of sequences used for phylogenetic analysis of the LOTUS domain. (**D**) List of sequences used for phylogenetic analysis of the OSK domain.

- Transparent reporting form

### Data availability

All data are available in the main text or the supplementary materials.

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
