## [Decision Letter]

**Acceptance summary:**

*oskar* is a master regulatory gene involved in insect early development that is only found in holometabolan insects. The gene's functions are well studied but its origins remain unknown. This study provides convincing phylogenetic evidence that the OSK domain of the protein encoded by the *oskar* gene appears to have been originally acquired from a bacterium. The study sheds significant light on the origin of a gene that is uniquely present in holometabolan insects and that plays a major role in insect early development.

**Decision letter after peer review:**

Thank you for submitting your article "Bacterial contribution to genesis of the novel germ line determinant *oskar*" for consideration by *eLife*. Your article has been reviewed by three peer reviewers, one of whom is a member of our Board of Reviewing Editors, and the evaluation has been overseen by Patricia Wittkopp as the Senior Editor. The following individual involved in review of your submission has agreed to reveal their identity: Eve Gazave (Reviewer #2).

The reviewers have discussed the reviews with one another and the Reviewing Editor has drafted this decision to help you prepare a revised submission.

Summary:

In this short report submitted to *ELife*, Blondel and collaborators aim at understanding the evolutionary history of a crucial gene for reproduction in some insects, *oskar*. This gene, not found outside this group of organisms, plays a sufficient and mandatory role for germ line determination. So far, the evolutionary history of this enigmatic gene was unknown. In this study, the authors conduct a suite of molecular evolutionary and bioinformatic analyses to unravel the origin of this important gene. They notably performed parametric and phylogenetic methods to detect potential HGT that has been suggested to be at the origin of Oskar. Based on these analyses and findings, they argue:

i) That one of the two domains of the Oskar protein, OSK, is more related to bacterial sequences while the other one, LOTUS, is clustering with eukaryotic sequences.

ii) That LOTUS is related to eukaryotes and that OSK clusters within bacterial sequences, more specifically with GDSL-like hydrolases, thanks to phylogenetic analyses of the two domains separately, as well as topology-constraint tests.

iii) That sequence characteristics, notably the GC3 content and the codon use, through the cosine distance analysis, are very different from the LOTUS and OSK domains which is consistent with the hypothesis of distinct origins for the two domains composing the Oskar gene.

Essential revisions:

The essential revisions fall into two major categories: concerns about the phylogenetic analyses and concerns about the codon usage analyses. The concerns about the codon usage analyses are quite substantial. Here, the authors have two options – one would be to remove completely this part from the manuscript (given that the HGT inference can stand on its own). The other would be to address the reviewers' concerns, which may reveal additional interesting biology.

Concerns about codon usage analyses:

Main text, twelfth paragraph: The reviewers were surprised that you were able to detect codon usage differences for HGT events that occurred that far back in time (e.g., third position synonymous sites are saturated between *D. melanogaster* and *D. pseudoobscura*). Thus, explanations of any differences between LOTUS and OSK parts of the *oskar* gene should be carefully scrutinized. In particular:

1) If we understand the methods correctly, the process used to calculate codon usage is somewhat unorthodox. By calculating the frequency of all 64 codons, rather than usage of alternate synonymous codons, this metric of similarity will be influenced by amino acid composition. Thus, genes where the 5' and 3' ends look quite different in their codon usage could result from very different amino acid composition between the two halves, rather than differences in synonymous codon usage. This raises the possibility that the large differences in codon usage between the LOTUS and OSK domains could be driven by differences in amino acid composition. Do either of these domains (especially OSK) have a particularly unusual or distinctive amino acid composition, relative to other arthropod genes?

2) We recommend that you conduct this analysis using a metric of codon usage that is independent of amino acid composition, such as the codon adaptation index or relative synonymous codon usage.

3) There are many other possible explanations for why the coding sequences of two domains in a protein may exhibit different codon usage that may have to do with domain folding, translational selection, etc. that would be worth considering as alternative explanations.

4) If the *oskar* gene is cut in half following the procedure used for the genes in the 17 genomes, rather than comparing just the LOTUS and OSK domains, are the 5' and 3' halves still significantly discordant for AT3/GC3/codon usage?

5) The manuscript states: "if evolutionary time had not completely erased the original GC3 content and codon use signatures from the putative bacterially donated sequence (OSK), we might detect some differences in these parameters both from the LOTUS domain, and from the host genome." All of the analyses presented appear to address the first issue (OSK vs. LOTUS), but I did not see any analyses addressing the second question, whether these parameters systematically differ between the OSK domain and the rest of the host genome. If some of the analyses in the paper do bear on this question, this connection should be made more explicit. This seems like an important question, and would help to inform whether the observed differences between the LOTUS and OSK domains are in fact due to unusual features of the OSK domain specifically.

6) In Figure 3—figure supplement 1, why is the correlation for AT3 between the 5' and 3' ends of genes positive, but negative for GC3? This seems counter-intuitive to me.

7) We don't understand this sentence: "Thus, we sampled each codon at least twice, preserving the coding frame." Why does cutting each gene in half at a randomly chosen location sample each codon twice (or more)?

8) Barplots in Figure 3A and B: it would be more informative to see the full distribution of correlations for all non-*oskar* genes. It would also be useful to see the points for the LOTUS and OSK domains highlighted in Figure 3—figure supplement 1. In general, Figure 3—figure supplement 1 appears to be more informative than Figure 3.

9) Figure 3C: we expect there are many points within the "intra-gene" set that are not displayed? If this is true, it would be more transparent to allow the boxplot whiskers to extend to the full range of the data.

10) Figure 3D is hard to understand – those two boxplots show two null distributions, but we don't see where the differences between the LOTUS and OSK domains are indicated.

Concerns about phylogenetic analyses/results:

11) Although the reviewers appreciate that the MUSCLE sequence alignment program is one of the state-of-the-art software for this type of analyses, it is fair to say that the use of different alignment programs can sometimes generate considerable variation in phylogenetic inference (e.g., https://academic.oup.com/mbe/article/30/3/642/1038709). We request that the authors use at least a couple of additional programs for sequence alignment (e.g., PRANK and T-Coffee) and rebuild the trees for the two domains. This should provide a sanity check that the key conclusions of their work are not sensitive to method/alignment software.

12) Main text, seventh paragraph: to really claim that the dipteran LOTUS domain sequences do not group together with the LOTUS domain sequences from other insects, you should do a topology constraint analysis (like you've done for the HGT). Is the ML topology where these two groups of sequences are forced into monophyly significantly different from the unconstrained ML topology? The reviewers think you need to show that these two topologies are significantly different to be able to claim that this is not simply an artifact of doing phylogenetic analyses using a short sequence alignment. Also note that your statement, "the phylogenetic interrelationships of these [LOTUS] sequences is largely consistent with the current species or family level trees for the corresponding insects", contradicts your findings/arguments in the aforementioned paragraph.

13) It is not clear how the two domains fused and when (approximately) they did so. I think it would be useful for the readers if the authors speculated in a small paragraph their hypothesis (based on their knowledge of the distribution of these two domains in insects) when the fused *oskar* gene originated.

Other broad comments:

14) Main text, fourth paragraph: why are you using two different thresholds for your blast searches of the protein versus the two domains?

15) The discussion and cited studies on HGT were too narrow and well known examples of bacterial to insect HGT not cited (e.g., Husnik et al., 2013; Sloan et al., 2014; Acuna et al., 2012, to mention just a few).

[Editors' note: further revisions were suggested prior to acceptance, as described below.]

Thank you for submitting your article "Bacterial contribution to genesis of the novel germ line determinant *oskar*" for consideration by *eLife*. Your article has been reviewed by three peer reviewers, one of whom is a member of our Board of Reviewing Editors, and the evaluation has been overseen by a Patricia Wittkopp as the Senior Editor. The following individual involved in review of your submission has agreed to reveal their identity: Eve Gazave (Reviewer #2).

The reviewers have discussed the reviews with one another and the Reviewing Editor has drafted this decision to help you prepare a revised submission.

Summary:

The study by Blondel et al. provides phylogenetic evidence that the OSK domain of the key developmental gene *oskar* appears to have been originally acquired from a bacterium. *oskar* is a key developmental gene and its functions are well studied but its origins remain unknown. This study sheds significant light on its origin and believe will be of interest not just to evo-devo afficionados, but more broadly to developmental biologists and evolutionary biologists. The manuscript is overall clearly written and straightforward to understand.

Essential revisions:

While the reviewers appreciated the authors' additional analyses on codon usage between the two domains of the *oskar* gene, they raised numerous concerns about the statistical analyses performed as well as for the inference that the observed patterns of codon usage may stem from the HGT event (see detailed comments below). The reviewers thought that the argument that the observed patterns are remnants of the HGT event is highly unlikely and hard to reconcile with everything that we know about the (fast) rate with which codon bias evolves. The reviewers also raised several remaining issues in the calculations of the codon usage statistics, which are substantial and would require additional, extensive analyses.

Given that the manuscript is a short communication and that the codon bias analyses are not mentioned in the title or Abstract (so their removal does not impact the manuscript's key result and discovery), our recommendation is that the authors remove all analyses associated with codon usage. If the authors can do so, there is broad agreement that the rest of this manuscript would be suitable for publication and would be accepted without the need for additional revisions (everyone concurred that the phylogenetic aspects of the manuscript are well done and will be interesting to a broad audience). Alternatively, if the authors feel strongly that they wish to keep the codon usage analyses in their manuscript, our recommendation is that the manuscript should be rejected. We sincerely hope that you decide to implement the reviewers' suggestions so that your manuscript can be accepted for publication at *eLife*.

Specific comments about codon usage analyses:

The reviewers appreciate the additional analyses that the authors have included regarding the differences in codon useage between the LOTUS and OSK domains of *oskar*. However, some of the changes are not presented clearly, and the rest appear to undermine the authors' claim that codon useage in *oskar* reflects its HGT hybrid origin.

First, there are some ambiguities and contradictions within the manuscript about the new analyses. The legend to Figure 3A states "Scatter plot representing the full distribution of correlations from Pearson correlation analysis for AT3 use in 3' and 5' halves". However, what is shown in Figure 3A is a bivariate plot – unless we misunderstand, calculating the correlation coefficient between the AT3 content for the 5' and 3' halves of each gene should yield a univariate distribution (one correlation coefficient for each gene) – and according to the axis labels, what is shown are the residuals (?) or Z-score (are these the same thing?) of AT3 for each 5' and 3' half of each gene. We are similarly confused about this statement: "The Z score ranges of correlations for the *oskar* domains are within the distribution for all genes in the genome, although correlations of use across the gene are lower for *oskar* than for non-*oskar* genes".

It is stated in the main text and in the Materials and methods that the residuals/Z-scores are calculated for each insect genome separately: "Pooling the residuals together revealed that the GC3 content was significantly different between the LOTUS and OSK domains, compared to what would be expected within an average gene in that genome"; "Then, Z scores for each sequence from the Intra-Gene, OSK or LOTUS domain sequences were calculated against the corresponding genome frequency distribution." But if this were the case, wouldn't the plots in Figure 3—figure supplements 1 and 2 be centered on (0,0)? At least Figure 3—figure supplement 1 looks as if the Z-scores were calculated from the combined distributions of AT3 pooled across all 17 genomes, which we do not think is appropriate, since genomes differ in AT/GC content. For this reason, we also think Figure 3A should be changed to Z-scores calculated for each genome separately.

Second, comparing the correlation in AT3/GC3 between the LOTUS and OSKAR domains with the average intra-gene correlation (either within a genome or across all genomes) does not seem to be a good indication of whether the correlation for *oskar* is unusual – as suggested by Figure 3A, many genes have correlations well above or below the mean for that genome, and likely have a lower correlation than observed for *oskar*. If even some genes in these genomes have AT3/GC3 correlations between their 5' and 3' halves that are as low or lower than *oskar's* (are some even negative?), and presumably this is not due to the same hybrid evolutionary origin as hypothesized for *oskar* , then how confident can we be that this low correlation reflects the HGT origin for *oskar*? We apologize for not raising this issue in the first round of review, but we did not quite understand what was represented in Figure 3C, D, E.

Third, we still think that the methods used to compare codon usage between the 5' and 3' ends of non-*oskar* genes are not fully comparable to comparing codon usage between the LOTUS and OSK domains. The OSK domain comprises the 3' 33% of *oskar* while the LOTUS domain comprises 16% of the protein, from 23-39% of the length from the 5' end. So the LOTUS domain makes up a smaller fraction of *oskar* than the methods used to sample the 5' halves of other genes, which could generate a more biased usage in the LOTUS domain. Probably more importantly, assuming that the LOTUS and OSKAR domains are functional, coherent units within the *oskar* protein, we might expect their amino acid composition to be unrepresentative of simply the 5' or 3' halves of genes, if most of those halves include multiple protein domains and/or truncate protein domains. It's hard to know what the perfect control set is here, but it would be informative to at least compare codon usage across two different discrete protein domains within a number of genes as was done for *oskar*.

Fourth, as the revised manuscript states, and as is clearly shown in Figures 3A, B (our questions about those figures notwithstanding), Figure 3—figure supplement 2, and Figure 3—figure supplement 6: codon useage in *oskar* genes is actually not unusual relative to their host genomes, or when it is unusual, this is true for both OSKAR and LOTUS domains (Figure 3—figure supplement 6A). Drawing the alternate conclusion in the face of these results confuses the manuscript.

Finally, we reiterate that we have difficulty believing, a priori, that "evolutionary time had not completely erased the original GC3 content and codon use signatures from the putative bacterially donated sequence (OSK)". Codon usage evolves over very rapid evolutionary timescales, (for example Akashi 1996, Genetics 144:1297), and can diverge genome-wide within a genus (see the 12 *Drosophila* genomes paper: Clark et al. 2007, Nature 450: 203-218, Figure 5). There is still much we do not understand about the evolution of synonymous codon usage (for example https://doi.org/10.1534/genetics.119.302542). Nonetheless, everything we do know suggests that either neutral or selective pressures should have long ago erased any codon usage signatures retained from a bacterial origin of the OSKAR domain. If the difference in AT3/GC3/codon useage is indeed unusual between the OSKAR and LOTUS domains of *oskar* (a conclusion that the manuscript has yet to convincingly demonstrate), then we insist that this is much more likely to be due to some unknown feature of *oskar* function that constrains its codon usage, than it is to reflect its evolutionary origin. As the authors mention in the revised manuscript, given that *oskar* mRNA is localized in the oocyte, this provides an entirely plausible constraint on the mRNA sequence independent of its coding role. We note that FlyBase also has annotated a lncRNA that almost entirely overlaps with the *oskar* mRNA in *D. melanogaster*; we don't know what evidence supports this annotation, but if true, this could indicate additional constraints that could shape codon usage. But in the end, we think such speculation is of limited value to the manuscript if the reader is not convinced that the codon usage is indeed unusually different between the two domains in *oskar*.

---

## [Author Response]

Essential revisions:The essential revisions fall into two major categories: concerns about the phylogenetic analyses and concerns about the codon usage analyses. The concerns about the codon usage analyses are quite substantial. Here, the authors have two options – one would be to remove completely this part from the manuscript (given that the HGT inference can stand on its own). The other would be to address the reviewers' concerns, which may reveal additional interesting biology.

We thank the reviewers for these suggestions. We chose to take the second of these two suggested options, and to address the reviewer’s concerns, as described for each point below, rather than completely removing the codon use analyses from the manuscript. Taken together, the results of the multiple new analyses that we performed supported our major conclusions. In some cases, notably for the phylogenetic analyses, they provided even stronger support for our hypothesis than in our original submission (Comments 11 through 13 below). In others, in particular for the codon use analyses, they indeed revealed additional nuances to the likely evolutionary history of the oskar gene (Comments 1 through 10 below). Overall we feel that the revised manuscript is stronger as a result of these revisions.

Concerns about codon usage analyses:Main text, twelfth paragraph: The reviewers were surprised that you were able to detect codon usage differences for HGT events that occurred that far back in time (e.g., third position synonymous sites are saturated between *D. melanogaster* and *D. pseudoobscura*). Thus, explanations of any differences between LOTUS and OSK parts of the oskar gene should be carefully scrutinized. In particular:

Summary response to concerns about codon use analysis: In response to these points about our codon use analyses, we have performed a number of new analyses of codon use, modified relevant figures (Figure 3), added new relevant supplementary figures (Figure 3—figure supplements 1-6), and have rewritten the section describing these analyses within the main text.

The codon use analysis in the revised manuscript now includes an assessment of the following four aspects of codon use:

1) GC3 and AT3 use: this analysis was included in the original manuscript, but we have added an new analysis as per the suggestion in Comment 4 below; results are shown in revised Figure 3, new Figure 3—figure supplement 1 and new Figure 3—figure supplement 2.

2) Comparison of codon adaptation index (CAI) between full length oskar, OSK and LOTUS domains, and all full length genes in a genome, as per Comment 2 below; results are shown within this document in Author response image 2.

3) Comparison of the difference in CAI between OSK and LOTUS domains (”δ CAI”), compared with the difference in CAI between the 5’ and 3’ ends of an average gene in the genome; results are shown within this document in Author response image 3.

4) Analysis of the CAI values for OSK and LOTUS domains in the context of the range of CAI values for all 3’ and 5’ halves of all genes in the genome; results are shown within this document in Author response image 4.

In the original manuscript, we had performed only analysis of parameter (1) above. In response to reviewer suggestions (Comment 2), we performed new analyses of parameters (2) through (4). We also refined our analysis of (1) relative to the original manuscript, by adding a second GC3/AT3 analysis where we changed the way we cut genes in half (see Comment 4). This analysis yielded the same result as our original analysis; indeed, we observed a slightly larger difference between the OSK and LOTUS domains with this method, than we had with our original method.

Taken together, analysis of these four parameters showed that the biggest difference in codon use between the OSK domain and the LOTUS domain, and also between the OSK domain and the genome as a whole, was that the difference in GC3 use between the OSK and the LOTUS domains was greater than the difference in GC3 use between the 5’ and 3’ halves of the average gene in the genome. Thus, in the revised manuscript, we retained original Figures 3A-C (with presentation modifications for Figure 3A as per Comment 8), which present the results of analysis (1) above. The results of new CAI analyses (done in response to Comments 2 and 5) are presented in new Figure 3—figure supplement 5.

Please see point by point responses to each of the ten comments regarding codon use analyses, in the following section:

1) If we understand the methods correctly, the process used to calculate codon usage is somewhat unorthodox. By calculating the frequency of all 64 codons, rather than usage of alternate synonymous codons, this metric of similarity will be influenced by amino acid composition. Thus, genes where the 5' and 3' ends look quite different in their codon usage could result from very different amino acid composition between the two halves, rather than differences in synonymous codon usage. This raises the possibility that the large differences in codon usage between the LOTUS and OSK domains could be driven by differences in amino acid composition. Do either of these domains (especially OSK) have a particularly unusual or distinctive amino acid composition, relative to other arthropod genes?

We take this excellent point by the reviewer. In response to this concern, we examined the relationship between the cosine difference in codon use between the two domains, and amino acid frequency cosine difference between the two domains. The results of this analysis are shown in Author response image 1. We found that, as per the reviewer’s suspicion, these two parameters were indeed highly correlated, such that we could not tell if this apparent different codon use result (based on cosine difference in codon use) was a consequence of different amino acid use or a difference in synonymous codon use. We therefore removed original Figure 3D from the revised manuscript.

2) We recommend that you conduct this analysis using a metric of codon usage that is independent of amino acid composition, such as the codon adaptation index or relative synonymous codon usage.

We appreciate this suggestion by the reviewer and agree that it would be ideal to be able to assess codon use with a metric that is independent of amino acid composition. For example, to calculate the codon adaptation index (CAI) for a gene, we would wish to use a well annotated set of highly expressed genes that display optimal codon use, based on genome and transcriptome analysis. However, to our knowledge, no such set of genes exists for most of the species studied here. We therefore took the following approach: for the 17 species that we used for the codon use analyses (listed in Supplementary file 1E), we calculated the frequency of codon use for every gene in the available transcriptomes, and calculated a “global CAI” for all genes in each of these species (rather than only for highly expressed genes, since we cannot be confident which ones those are based on the available data for these species). We also calculated the codon use frequencies for the full length oskar gene, and for the two domains of oskar, LOTUS and OSK.

Using these frequencies, we performed two analyses. First, for each oskar ortholog in each of these 17 species, we compared the codon use in the full length gene, and in each of its two domains, to the codon use of all genes in the genome. We found that codon use for full length oskar, and for each of its two domains LOTUS and OSK, was well within the distribution of codon use for all genes in the genome (Author response image 2).

**Author response image 2. respfig2:** 

Second, we compared the difference in codon use between the LOTUS and OSK, to the difference in codon use between the 3’ and 3’ halves of all genes in the genome, and again found that the values were well within the distribution of 5’ vs. 3’ codon use for all genes in the genome (Author response image 3).

**Author response image 3. respfig3:** 

Thus, this analysis does not reveal any significant difference in codon use measured in this way, between the OSK and LOTUS domains of these insect oskar orthologs. The revised manuscript does include some new CAI analyses examining possible differences between OSK domain codon use and codon use in the rest of the genome (discussed below in Comment 5 Response). However, we include Author response image 2 and 3, as it is not a useful way to compare codon use between the OSK and LOTUS domains, given the likely confounding effect of distinct amino acid use between domains that the reviewer helpfully suggested we consider (Author response image 1).

3) There are many other possible explanations for why the coding sequences of two domains in a protein may exhibit different codon usage that may have to do with domain folding, translational selection, etc. that would be worth considering as alternative explanations.

We take this point by the reviewer. In response, we have added new text to the revised manuscript that discusses these alternative explanations for the observed differences in codon usage between the two domains (main text).

4) If the oskar gene is cut in half following the procedure used for the genes in the 17 genomes, rather than comparing just the LOTUS and OSK domains, are the 5' and 3' halves still significantly discordant for AT3/GC3/codon usage?

This is an interesting suggestion by the reviewer. In response to this query, we performed a new analysis in order to assess whether cutting the oskar gene following the same procedure as devised for the Intra-Gene null distribution would change the overall results in codon use. We cut each oskar mRNA transcript in half at a random location while preserving the coding frame, in the same manner as that described for the Intra-Gene distribution analysis in the original manuscript. We then ran the same GC3 AT3 and wobble positions analyses on those “half sequences” of oskar orthologues. We found that the overall difference in codon usage between the two halves was the same as in our original comparison of just the LOTUS and OSK domains. In other words, the two domains were uncorrelated in their codon use, in contrast to most genes in the genome, which did show a correlation in codon use between the two halves of a gene. The only difference we noted relative to our original analysis was in the T3 wobble position which showed some correlation between OSK and LOTUS domains in our original analysis, but showed no correlation at all in the new random cut analysis. Thus, overall the main finding of discordant AT3/GC3 codon use between the two domains of oskar remains unchanged by this new analysis. We have included the results of the new random cut analysis in new Figure 3—figure supplement 4.

5) The manuscript states: "if evolutionary time had not completely erased the original GC3 content and codon use signatures from the putative bacterially donated sequence (OSK), we might detect some differences in these parameters both from the LOTUS domain, and from the host genome." All of the analyses presented appear to address the first issue (OSK vs. LOTUS), but I did not see any analyses addressing the second question, whether these parameters systematically differ between the OSK domain and the rest of the host genome. If some of the analyses in the paper do bear on this question, this connection should be made more explicit. This seems like an important question, and would help to inform whether the observed differences between the LOTUS and OSK domains are in fact due to unusual features of the OSK domain specifically.

We thank the reviewer for this important point. The reviewer is correct in that we did not explicitly test differences in codon use parameters between the OSK domain and the rest of the genome in the original manuscript. To address this issue, we made use of the CAI approach as explained in the response to Comment 2. We calculated the CAI for each transcript in each of these 17 genomes, for the full length oskar mRNA, and for the LOTUS and OSK domains. We found that the latter three CAI values fell well within the ranges of the genome-wide CAI distribution, for all 17 genomes studied (new Figure 3—figure supplement 6A).We then asked if the difference in CAI between the LOTUS and OSK domain was larger than what would be expected than the differences between the 3’ and 5’ end (cut in ”half” according to the procedure described in response to Comment 4) of an “average gene” in each genome. Using the Intra-Gene Distribution as a null model, we computed the CAI difference between the 3’ and 5’ end of randomly cut transcripts, and did the same between the LOTUS and OSK sequences. We found no significant difference between the CAI difference for the LOTUS and OSK domains, and the distribution of CAI differences for other genes in the genome (new Figure 3—figure supplement 6B).

Finally, we asked how the LOTUS and OSK domain CAI values compared to the median genome-wide CAI values for the 5’ and 3’ ends of each of the other genes in a genome. To address this, we computed a CAI Z score for all genes as follows: for each genome we calculated the mean CAI and the standard deviation of the CAI for each of the 5’ and 3’ parts of the transcripts of all predicted genes in each of the 17 genomes under study. Using these values, we computed (a) the Z score relative to the global CAI for that genome, for each of the 5’ and 3’ halves, and (B) the Z score for each of the LOTUS and OSK domains of the oskar ortholog of each genome. We found that the Z score for the LOTUS and OSK domains fell within the distribution of Z scores other genes in the genome (new Figure 3—figure supplement 6C).

Thus, based on these CAI analyses, we did not detect any unusual codon use features of the OSK domain relative to the remainder of the genome. We include and discuss the results of these new analyses in the revised manuscripts as Figure 3—figure supplement 6.

6) In Figure 3—figure supplement 1, why is the correlation for AT3 between the 5' and 3' ends of genes positive, but negative for GC3? This seems counter-intuitive to me.

We found that the aggregation of the data from all 17 genomes analyzed here generated a distribution that appeared negatively correlated. Indeed, we agree with the reviewer that this seems counterintuitive. However, when each genome is observed individually, we can see that correlations are positive. To clarify this for the reader, we have added two new supplementary figures showing the individual correlations for GC3 and AT3 for each genome analyzed (Figure 3—figure supplements 1 and 2).

7) We don't understand this sentence: "Thus, we sampled each codon at least twice, preserving the coding frame." Why does cutting each gene in half at a randomly chosen location sample each codon twice (or more)?

This comment from the reviewer made us realize that we should clarify our description of this analysis. To this end, we have added a new schematic to diagram our analysis method (Figure 3—figure supplement 5), and have rewritten the corresponding text (subsection “Generation of Intra-Gene Distribution of Codon Use”).

8) Barplots in Figure 3A and 3B: it would be more informative to see the full distribution of correlations for all non-oskar genes. It would also be useful to see the points for the LOTUS and OSK domains highlighted in Figure 3—figure supplement 1. In general, Figure 3—figure supplement 1 appears to be more informative than Figure 3.

We agree with this point by the reviewer. In response, we have made the following changes:

a) We have added new representations of the plots shown in the original Figure 3—figure supplement 1 (now called Figure 3—figure supplement 2 in the revised manuscript). These new representations now appear as revised Figure 3, new panels A and B. As suggested, these plots identify the 17 oskar genes in the analysis (red dots) so that the reader can see how their Z scores compare with those of non-oskar genes.

b) Because the new panels Figure 3A and B show the correlations for every gene analyzed in every one of the 17 genomes included in the analysis, they are very densely populated. We therefore wished to include an easier way for readers to visualize how the GC3 correlations for the LOTUS and OSK domains of oskar, compare with the GC3 correlations for the 5’ and 3’ parts of non-oskar genes. To this end, we added two new supplementary figures (Figure 3—figure supplements 1 and 2 in the revised manuscript) that provide a contour map for the scatter plot of the GC3 and AT3 correlations for the 5’ and 3’ parts of each of the 17 genomes included in this analysis, and shows where the LOTUS/OSK domain G3 correlation for each oskar gene falls relative to all non-oskar genes in each genome (red dot).

c) Because we believe that it is easier to compare the residuals in the original bar plots that we showed in the original Figure 3A and B, we have retained these in the revised Figure 3, as panels C and D.

9) Figure 3C: we expect there are many points within the "intra-gene" set that are not displayed? If this is true, it would be more transparent to allow the boxplot whiskers to extend to the full range of the data.

We take this point by the reviewer, and in response, have modified Figure 3C to show the full range of the data.

10) Figure 3D is hard to understand – those two boxplots show two null distributions, but we don't see where the differences between the LOTUS and OSK domains are indicated.

As Comment #1 above helped us realize, our choice of distance metric in this analysis could not exclude the possibility that differences in codon use between the two domains of oskar were due to differences in amino acid use. We have therefore eliminated this panel from the revised Figure 3. Please see our responses to Comments #1 through 9 for further explanations of the rationale behind this revision.

Concerns about phylogenetic analyses / results:11) Although the reviewers appreciate that the MUSCLE sequence alignment program is one of the state-of-the-art software for this type of analyses, it is fair to say that the use of different alignment programs can sometimes generate considerable variation in phylogenetic inference (e.g., https://academic.oup.com/mbe/article/30/3/642/1038709). We request that the authors use at least a couple of additional programs for sequence alignment (e.g., PRANK and T-Coffee) and rebuild the trees for the two domains. This should provide a sanity check that the key conclusions of their work are not sensitive to method/alignment software.

This is an excellent suggestion by the reviewer. In response to this comment, we created new alignments using PRANK and T-Coffee for both domains, and trimmed these alignments using the same parameters as described in the original manuscript for the MUSCLE alignments. The new PRANK and T-Coffee alignments are provided in the revised supplementary files (OSK_prank_aligned.fasta and OSK_tcoffee_aligned.fasta). We then used these new PRANK and T-Coffee alignments to rebuild both trees using RAXML with the same parameters described in the original manuscript.

For the OSK domain, as did the original MUSCLE-based tree, the topology of the PRANK-based and T-Coffee-based trees revealed the OSK domain as more closely related to bacterial than to eukaryotic sequences. The following minor differences in topology between the trees did not affect this bacterial (rather than eukaryotic) affinity relationship of the OSK domain:

PRANK Tree

1) The closest bacterial nodes to the OSK domain formed a polytomy in the original MUSCLE-based tree, but are resolved in the PRANK-based tree. Specifically, OSK now branches within the group of bacteria that it was previously sister to.

2) The support values are globally higher in the PRANK-based tree in comparison to the previous MUSCLE-based tree.

3) Small differences in the topology of the bacterial sequences are seen throughout the PRANK-based tree. Specifically, a monophyletic group of Bacteroidetes changed its branch position from the base of the tree to within the main Firmicutes monophyletic group, as a sister group to the CAZ3 fungal sequences. Two Bacteroidetes sequences (R7ADB4) and one from an environmental sample branch at the base of a large clade of Firmicutes, rather than within that clade as they did in the MUSCLE-based tree. Finally, the Proteobacteria sequences now branch outside of the larger Firmicutes clade.

T coffee Tree

1) Two Streptococcus sequences now branch with the bacteria clade sister to OSK sequences.

2) Similar to the PRANK-based tree, the CAZ 3 sequences are now more closely related to the Bacteroidetes clade, though they are not sister to this clade unlike in the PRANK-based tree. Instead, the CAZ3 sequences branch as an outgroup of the main clade containing OSK and the Firmicutes.

3) Similar to the PRANK tree, The Proteobacteria clade branches outside the main group of Firmicutes bacterial sequences.For the LOTUS domain, the PRANK-based and T-Coffee-based trees showed that, as for the MUSCLE-based tree presented in the original manuscript, the closest relatives of the LOTUS domain of Osk proteins are other LOTUS domains found in eukaryotic proteins, as would be expected for a gene of animal origin. In trees made with all of these alignment methods, the phylogenetic interrelationships of these sequences is largely consistent with the current species or family level trees for the corresponding insects.

The following specific minor differences, none of which affect these major conclusions, were observed between trees:

PRANK Tree

1) In the PRANK-based tree, the dipteran oskar LOTUS domains grouped within a common node with the TUDOR 7 LOTUS domain and the other insect oskar LOTUS domains, and a single arthropod Tudor 5 LOTUS domain (A0A0J7KVQ7) branched sister to the dipteran oskar LOTUS domains

2) All the arthropod LOTUS domains are grouped under one common node (with the MUSCLE tree they were split between the TUD5 and TUD7 groups).

3) Other differences in the topology of the non-oskar sequences are seen in the PRANK-based tree. Notably, the internal branching of mammalian TUD5 sequences changed to form two main clades, one containing mostly primate sequences, the other containing non-primate TUD5 sequences. In addition, the internal branching of arthropod TUD7 sequences changed, such that arthropod sequence relationships followed order relationships, including with three hymenopteran, two hemipteran and two coleopteran sequences.

T Coffee tree

1) The Oskar LOTUS domain sequences formed a monophyletic clade that is sister to the TUD7 sequences.

2) TUD5 sequences showed similar relationships as those recovered in the MUSCLE Tree. Similar to the PRANK-based tree, the primate TUD5 sequences formed a monophyletic group.

3) TUD7 sequences formed two main clade, one composed of sequences from Chondrichthyes, and one composed of mammalian sequences.

In the revised manuscript, we present the new PRANK-based trees in new Figure 2—figure supplement 6 (LOTUS) and Figure 2—figure supplement 7 (OSK), and the new T-Coffee-based trees in new Figure 2—figure supplement 10 (LOTUS) and Figure 2—figure supplement 11 (OSK). We also present graphic visualizations of the similarities and differences between the trees generated with the different alignment methods (Figure 2—figure supplement 8 (MUSCLE vs. PRANK: OSK), Figure 2—figure supplement 9 (MUSCLE vs. PRANK :LOTUS), Figure 2—figure supplement 12 (MUSCLE vs. T-Coffee: OSK) and Figure 2—figure supplement 13 (MUSCLE vs. T-Coffee: LOTUS)).

12) Main text, seventh paragraph: to really claim that the dipteran LOTUS domain sequences do not group together with the LOTUS domain sequences from other insects, you should do a topology constraint analysis (like you've done for the HGT). Is the ML topology where these two groups of sequences are forced into monophyly significantly different from the unconstrained ML topology? The reviewers think you need to show that these two topologies are significantly different to be able to claim that this is not simply an artifact of doing phylogenetic analyses using a short sequence alignment. Also note that your statement, "the phylogenetic interrelationships of these [LOTUS] sequences is largely consistent with the current species or family level trees for the corresponding insects", contradicts your findings/arguments in the aforementioned paragraph.

This is an excellent suggestion by the reviewers. In response to this suggestion, we used SOWHAT to compare the probabilities of trees with various topology constraints. We found that the ML topology requiring monophyly of all insect LOTUS domains was significantly less likely than the unconstrained topology, with a p-value of 0.037. We present the results of this analysis in new Figure 2—figure supplement 5.

13) It is not clear how the two domains fused and when (approximately) they did so. I think it would be useful for the readers if the authors speculated in a small paragraph their hypothesis (based on their knowledge of the distribution of these two domains in insects) when the fused oskar gene originated.

In response to this comment, we have added new text of the revised manuscript to explain why that we believe that this fusion event can be dated most precisely to at least the time before major insect divergence (main text).

Other broad comments:14) Main text, fourth paragraph: why are you using two different thresholds for your blast searches of the protein versus the two domains?

In the original manuscript we had done searches in November of 2015 using the full length protein as a query, applying an E-value threshold of 0.01 for matches to the whole sequence, and an E-value threshold of 10 for matches to the OSK and LOTUS domains. We chose these different thresholds because we wished to cast a wide net in our initial search for potential domains that might be related to the domains of oskar. To determine whether having the less stringent cutoff for the full length protein search would have changed our results, we re-did the search in May of 2019 using the *D. melanogaster* osk full length sequence as a query and imposing an E-value threshold of 10. This search yielded a similar result to the one described in the original manuscript: that is, all sequences were from an insect species, with the exception of three TUDOR 5 Pocillopora damicornis (coral) sequences with E-values of 1.8, 1.8 and 1.9. Those sequences had been added to the nr database in 2018 (https://www.ncbi.nlm.nih.gov/assembly/GCF_003704095.1) and thus were not present during our first search in 2015. Searches with hmmsearch and the OSK.hmm model revealed that none of these sequences contained a OSK domain, so they were not included in any further analyses. In response to this comment, we have clarified these points in the revised Materials and methods section. Subsection “BLAST searches of Oskar” is an updated BLAST section where we explain the choices described above. In addition, we have added a reference to the specific relevant Materials and methods section, with a comment about E-value choices, to the revised main text.

15) The discussion and cited studies on HGT were too narrow and well known examples of bacterial to insect HGT not cited (e.g., Husnik et al., 2013; Sloan et al., 2014; Acuna et al., 2012, to mention just a few).

In response to this comment, we discuss and include a broader range of examples of HGT studies from the literature in the revised manuscript, including the ones specifically suggested by the reviewer (main text).

[Editors' note: further revisions were suggested prior to acceptance, as described below.]

Essential revisions:While the reviewers appreciated the authors' additional analyses on codon usage between the two domains of the oskar gene, they raised numerous concerns about the statistical analyses performed as well as for the inference that the observed patterns of codon usage may stem from the HGT event (see detailed comments below). The reviewers thought that the argument that the observed patterns are remnants of the HGT event is highly unlikely and hard to reconcile with everything that we know about the (fast) rate with which codon bias evolves. The reviewers also raised several remaining issues in the calculations of the codon usage statistics, which are substantial and would require additional, extensive analyses.Given that the manuscript is a short communication and that the codon bias analyses are not mentioned in the title or Abstract (so their removal does not impact the manuscript's key result and discovery), our recommendation is that the authors remove all analyses associated with codon usage. If the authors can do so, there is broad agreement that the rest of this manuscript would be suitable for publication and would be accepted without the need for additional revisions (everyone concurred that the phylogenetic aspects of the manuscript are well done and will be interesting to a broad audience). Alternatively, if the authors feel strongly that they wish to keep the codon usage analyses in their manuscript, our recommendation is that the manuscript should be rejected. We sincerely hope that you decide to implement the reviewers' suggestions so that your manuscript can be accepted for publication at eLife.

We have removed all codon use analysis from the manuscript as suggested. Accordingly, we have not responded in a point by point fashion to each of the specific comments about codon use analyses.